

# Genotyping by sequencing for the construction of oil palm (*Elaeis guineensis* Jacq.) genetic linkage map and mapping of yield related quantitative trait loci

Fakhrur Razi Mohd Shaha[1,2], Pui Ling Liew[1],
Faridah Qamaruz Zaman[2], Rosimah Nulit[2], Jakim Barin[3],
Justina Rolland[3], Hui Yee Yong[1] and Soo Heong Boon[1]

[1] ACGT Sdn. Bhd. & Laboratories, Bukit Jalil, Kuala Lumpur, Malaysia
[2] Department of Biology, Faculty of Science, Universiti Putra Malaysia, Serdang, Selangor, Malaysia
[3] Wisma Pertanian Sabah, Department of Agriculture Sabah, Kota Kinabalu, Sabah, Malaysia

Corresponding author
Fakhrur Razi Mohd Shaha,
rfakhrur@genting.com,
rfakhrur@gentingplantations.com

## ABSTRACT

**Background:** Oil palm (*Elaeis guineensis* Jacq.) is one of the major oil-producing crops. Improving the quality and increasing the production yield of oil palm have been the primary focuses of both conventional and modern breeding approaches. However, the conventional breeding approach for oil palm is very challenging due to its longevity, which results in a long breeding cycle. Thus, the establishment of marker assisted selection (MAS) for oil palm breeding programs would speed up the breeding pipeline by generating new oil palm varieties that possess high commercial traits. With the decreasing cost of sequencing, Genotyping-by-sequencing (GBS) is currently feasible to many researchers and it provides a platform to accelerate the discovery of single nucleotide polymorphism (SNP) as well as insertion and deletion (InDel) markers for the construction of a genetic linkage map. A genetic linkage map facilitates the identification of significant DNA regions associated with the trait of interest *via* quantitative trait loci (QTL) analysis.

**Methods:** A mapping population of 112 $F_1$ individuals from a cross of Deli *dura* and Serdang *pisifera* was used in this study. GBS libraries were constructed using the double digestion method with *Hind*III and *Taq*I enzymes. Reduced representation libraries (RRL) of 112 $F_1$ progeny and their parents were sequenced and the reads were mapped against the *E. guineensis* reference genome. To construct the oil palm genetic linkage map, informative SNP and InDel markers were used to discover significant DNA regions associated with the traits of interest. The nine traits of interest in this study were fresh fruit bunch (FFB) yield, oil yield (OY), oil to bunch ratio (O/B), oil to dry mesocarp ratio (O/DM) ratio, oil to wet mesocarp ratio (O/WM), mesocarp to fruit ratio (M/F), kernel to fruit ratio (K/F), shell to fruit ratio (S/F), and fruit to bunch ratio (F/B).

**Results:** A total of 2.5 million SNP and 153,547 InDel markers were identified. However, only a subset of 5,278 markers comprising of 4,838 SNPs and 440 InDels were informative for the construction of a genetic linkage map. Sixteen linkage groups were produced, spanning 2,737.6 cM for the maternal map and 4,571.6 cM for the paternal map, with average marker densities of one marker per 2.9 cM and one per 2.0 cM respectively, were produced. A QTL analysis was performed on nine traits;

however, only QTL regions linked to M/F, K/F and S/F were declared to be significant. Of those QTLs were detected: two for M/F, four for K/F and one for S/F. These QTLs explained 18.1–25.6% of the phenotypic variance and were located near putative genes, such as casein kinase II and the zinc finger CCCH domain, which are involved in seed germination and growth. The identified QTL regions for M/F, K/F and S/F from this study could be applied in an oil palm breeding program and used to screen palms with desired traits *via* marker assisted selection (MAS).

## INTRODUCTION

Oil palm, *Elaeis guineensis* Jacq., is one of the main oil-producing crops and is considered the Golden Crop. There are two major commercialized species of oil palm *E. guineensis* and *E. oleifera*. The former originates from West Africa, while the latter is from South and Central America (*Corley & Tinker, 2016*). The planting of *E. guineensis* is favored over *E. oleifera* in Southeast Asia, due to its higher yield (*Singh et al., 2013*). There are three different types of oil palm fruit, *dura*, *pisifera* and *tenera*. The hybrid *tenera* has a higher oil yield compared to its *dura* and *pisifera* parents which results in an increased 30% yield compared to *dura* (*Corley, 2006*).

The breeding progress for yield improvement in major crops, however, is still at a low rate which is about 1–2% *per annum* (*Soh, 2004*; *OECD/FAO, 2019*). It has been reported that there has been an almost stagnant yield improvement of the oil palm in Malaysia for the past 20 years, from 2000 to 2022 (*Basiron, 2002*; *Parveez, 2023*). However, over the same time period, the yield improvement for annual crops such as soybean, rapeseed and sunflower has shown a substantial yield increment, between 6% and 62%, (*World Oil, 2019*). One of the limitations of the conventional oil palm breeding pipeline is its lifespan, which can be up to 40 years (*Kushairi et al., 2019*). The unstable genotyping and low hereditability may be observed during the early cycle, selecting plants with any trait improvement at later stage is much more challenging (*Soh, 2004*) as it would take decades to generate and release any new improved variety of oil palm.

Nowadays, mining DNA markers using the next generation sequencing (NGS) approach is more feasible and affordable for many institutes as the sequencing cost is reduced. This approach is able to discover genome wide markers. For oil palm, *Pootakham et al. (2015)* and *Bai et al. (2017)* reported that the construction of dense oil palm genetic linkage maps using the NGS platform would pave the way to QTL identification. Similar approaches were also found to be useful in various plant species; particularly sugarcane (*Balsalobre et al., 2017*), soybean (*Liu et al., 2017*), wheat (*Gao et al., 2015*), and alfalfa (*Zhang et al., 2019*). Leveraging this marker technology in the oil palm breeding program could not only reduce the time taken to produce a new variety, but could also possibly unlock oil palm yield improvement in the near future.

The selection of SNP as a molecular marker system is expected to increase the number of DNA markers obtained in this study because SNPs are the most abundant and widely distributed markers across genome (*Agarwal, Shrivastava & Padh, 2008*; *Giordano et al., 1999*). As a result, a dense genetic linkage map would be generated to facilitate the identification of QTL regions associated with the trait of interest. However, QTL analyses could be significantly affected by the trait's heritability. It is widely reported that FFB and oil yield are considered low hereditability traits that are highly influenced by the environment. Thus, the identification of QTL regions associated with those traits would be more challenging compared to the identification of fruit development traits.

This study was initiated to determine the DNA markers associated with oil palm's economical traits, mainly yield and its yield components, such as fresh fruit bunch (FFB) and oil yield (OY), as well as bunch quality characters, which include oil to bunch ratio (O/B), oil to dry mesocarp ratio (O/DM), oil to wet mesocarp ratio (O/WM), mesocarp to kernel ratio (M/F), kernel to fruit ratio (K/F), shell to fruit ratio (S/F), and fruit to bunch ratio (F/B) that can be found in an oil palm cross, Deli *dura* and Serdang *pisifera*. Many oil palm genetic linkage maps have been published (*Billotte et al., 2005*; *Singh et al., 2009*; *Ting et al., 2014*; *Lee et al., 2015*; *Pootakham et al., 2015*; *Bai et al., 2017*). However, none of the publications had reported using a similar cross similar to this study, Deli *dura* and Serdang *pisifera* on an NGS pipeline to discover polymorphic markers. Furthermore, progenies produced from different parental genetic sources could segregate and generate a genetic variation among the population. Thus, this study could discover new DNA regions in the oil palm's genome associated with oil yield related traits.

## MATERIALS AND METHODS

### Plant material and DNA extraction

Leaf tissue was collected from 112 palms and two parent palms. These 112 *tenera* full sibs were generated from a cross of Deli *dura* and a Serdang *pisifera*. The palms were managed and maintained by the Department of Agriculture (DOA) at a plantation in Quoin Hill, Tawau, Sabah, Malaysia (4°24′N, 118°1′E). The leaf samples were collected from individual trees and immediately submerged in liquid nitrogen (LN) before being stored at −80 °C freezer until DNA extraction was executed. One hundred milligrams of each individual leaf sample were used as a starting material for DNA extraction. The samples were frozen in the LN before being disrupted with beads using the TissueLyser II. The genomic DNA from the tissue samples was extracted using the DNeasy® Plant Mini Kit (Qiagen, Hilden, Germany), following the manufacturer's protocol, but with minor modifications to prolong the incubation time.

The genomic DNA concentration and purity were quantified and qualified using the Nanodrop-1000 spectrophotometer (Thermo Fisher Scientific, Waltham, MA, USA). The samples were normalized in 100 ng/μl using Buffer AE (Qiagen, Hilden, Germany). Then, the integrity of the extracted DNA was further examined on a 1% TAE agarose gel (Choice Care, Malaysia) for an hour at 100 V. The DNA ladder (Thermo Scientific, Waltham, MA, USA) was added to serve as an indicator for reference bands. After an hour, the agarose gel was stained in an ethidium bromide solution for half an hour before being

viewed under an AlphaImager HP DE-500 (Alpha Innotech, San Leandro, CA, USA). DNA samples of all 112 $F_1$ and the two parents that passed the quality control checks with acceptable ratios of absorbance reading at A260/A280 and A260/A230 1.8 and 2.0–2.2 (Thermo Scientific, Waltham, MA, USA), respectively, were considered to have acceptable DNA purity. These DNA samples were stored in a 4 °C chiller (Thermo Scientific, Waltham, MA, USA) prior to the sequencing works.

## GBS library construction

The sequencing works as well as the GBS library construction on the 112 progenies and their parents was outsourced to Genting Laboratory Service Sdn. Bhd. (GLS). In this study, the quality of genomic DNA was quantified using the Qubit® dsDNA HS Assay Kit (Life Technologies, Carlsbad, CA, USA). GBS libraries were constructed using the double digestion methods with *Hind*III and *Taq*I enzymes (New England Biolabs, Ipswich, MA, USA) according to the standard GBS protocol (*Elshire et al., 2011*). *Hind*III is a type II endonuclease that recognizes a degenerate 6 bp sequence (5′…AAGCTT…3′) and cleavages between the AA of the sequence resulting in 5′ overhang. *Taq*I. on the other hand, recognizes a degenerate 4 bp sequence (5′…TCGA…3′) and cuts between the TC of the sequence to create a 5′ overhang. A total of 200 ng of genomic DNA from each sample was digested with the enzymes for 2 h at 75 °C. The digested samples were examined through gel electrophoresis (Scie-Plas, Gloucestershire, England) on a 1% TAE agarose gel before the ligation adapters were added. The ligation products were pooled, and those with sizes of 400–600 bp were selected using Pipin Prep (Sage Science, Beverly, MA, USA). Two different types of adapters, barcode and common adapters, were used to enable paired-end and multiplex sequencing on the Illumina HiSeq2000 platform (Illumina Inc., San Diego, CA, USA).

In this experiment, 12 unique adapters with specific barcodes were designed to carry out the multiplexing of 12 samples per Illumina flow cell lane. The library construction was performed using a manufacturer's protocol. Finally, libraries without adapter dimers were retained for DNA sequencing using the Illumina HiSeq2000 sequencing platform following the standard protocol (Illumina Inc., San Diego, CA, USA).

## SNP and InDel calling

After the sequencing run, the sequencing reads were filtered using a FASTQ file with a quality confidence threshold of ≥25. The sequencing reads with the minimum read coverage of five for both parents and progenies were mapped and aligned to the EG5 genome build (*Singh et al., 2013*), which was obtained from the GenomeSawit webpage (http://genomsawit.mpob.gov.my), using the Burrows-Wheeler Aligner (*Li & Durbin, 2009*). Then, SNP and InDel callings were performed using the genome analysis toolkit, GATK ver. 3.3.0, with default parameters (*McKenna et al., 2010*).

## SNP and InDel filtering

Using sequence data with a minimum depth reading of 5X and a Q-value of 25 for each sample, markers across the progeny palms with less than 10% missing calls and major

alleles, as well as a genotype frequency of more than 95% were excluded. The filtering task was performed using VCFtools ver. 0.1.10 (*Danecek et al., 2011*) before the final versions of the SNP and InDel genotyping calls were tabulated into PLINK format (*Purcell et al., 2007*). In the preparation of the input file for the construction of a genetic linkage map, the allele calls for the parental palms were also investigated. Markers with genotypes that were different from the expected Mendelian segregation patterns between the progenies and the parents for $F_1$ segregation were filtered out.

## Genotype data preparation

Markers with major allele and genotype frequencies of more than 95% were eliminated using Microsoft Excel 2010. Subsequently, the heterozygous markers were coded based on JoinMap ver. 4.1 for a cross pollinated (CP) population. Markers coded with <lmxll> and <nnxnp> represent markers with one heterozygous parents, and <hkxhk>, <efxeg> and <abxcd> represent markers with both parents being heterozygous with the presence of two alleles, three alleles and four alleles, respectively.

The segregation ratios for each marker were examined using the chi-square goodness-of-fit test, and the threshold $p$-value was set at 0.001. Segregation distortion markers with a $p$-value of less than 0.001 were excluded. Subsequently, the marker similarities were checked for reducing marker redundancy, as two identical markers would be located at the same position in the linkage map. If a pair of markers has a similarity value that exactly equals 1, one of the markers in the pair would be excluded. After removing markers with the filtering parameters as described above, the data set was ready for the construction of genetic linkage maps.

## Genetic linkage map construction

Genetic linkage maps of the female parent, D200, and the male parent, QP447, were constructed using the significant SNP and InDel markers of the 112 $F_1$ population. The analysis was performed using the JoinMap 4.1 software (*Van Ooijen, 2006*) and the maps were drawn using the MapChart V2.2 software (*Voorrips, 2002*), as described below. Further data clean-up was performed prior to the construction of the genetic linkage map, as described in the following details. In order to capture the population structure in this study, 2,291 informative SNP markers were used to plot a principal component analysis (PCA) using PLINK ver. 1.90 (*Weeks, 2010*) and visualization using Microsoft Excel 2010 to access the $F_1$ mating design.

Maternal and paternal maps were constructed using the filtered markers sets. To determine the significant LOD threshold, the start value was set at 2.0 and the end value at 10.0 with a step size of 1.0 (*Van Ooijen, 2006*). The LOD score was calculated for the recombination frequency and was performed using the software based on the G2 statistic, G2 = 2 $\sum$ O log (O/E) where, O is the observed value and E is the expected number of individuals in a cell, log is the natural algorithm and $\sum$ is the sum over all cells. Then, the values were multiplied by 0.217, which is derived from 0.5*log10(e) to obtain the normal LOD score (*Van Ooijen, 2006*). The grouping parameter was selected at a LOD-value of more than 4.0 which indicates the likelihood of two loci being linked is greater than
10,000:1. The maximum likelihood (ML) mapping algorithm was selected to compute the mutual distance of loci in the respective groups.

The map order was further improved by eliminating markers exhibiting a Nearest Neighbor Stress value (N.N. stress) of more than 3 cM. The N.N. stress measurement assesses the extent to which markers are located outside of their expected region (*Van Ooijen, 2006*). After removing markers with a N.N. stress of more than 3 cM in each of the linkage groups, the maps were recalculated until the optimal map order was achieved. Finally, the genetic map distances were calculated using the Haldane mapping function. Once the linkage group maps were effectively constructed, both parental maps were drawn and visualized using MapChart 2.2 (*Voorrips, 2002*), and the genetic map information was used in the QTL analysis.

## Phenotypic traits and analyses

In this study, the FFB yield of the individual palm from five consecutive years (1984–1988) was recorded, and the average FFB yield was calculated. With a minimum of at least three bunches per palm, O/B, O/DM, O/WM, M/F, K/F, S/F and F/B were analyzed as per standard industry practice (*Blaak, Sparnaaij & Menedez, 1963*). The OY was calculated from the multiplication of the 5-year average FFB yield and the average O/B. A total of nine traits (Supplemental Data) was compiled for descriptive statistics. Out of a total of 112 genotyped palms, only 111 palms were collected for FFB yield and 97 palms with yield related components. The phenotype data collection was performed by the Department of Agriculture (DOA), Sabah. During the field data recording, a few palms died, but their leaves were already sampled earlier for DNA sequencing and molecular analysis. As a result, fourteen samples were excluded from the bunch analysis due to the insufficient number of bunches recorded. A Pearson correlation analysis and the significant level between traits were also computed using SPSS statistical software. The R-package heritability was used to estimate narrow-sense heritabilities for the nine traits (*Kruijer, 2019*).

## QTL analysis

To declare the presence of a significant QTL, the threshold LOD values were estimated at the genome-wide (GW) and chromosome-wide (CW) levels. For both cases, the acceptable error level of the permutation test with 1,000 iterations was 5%. The interval mapping (*Lander & Botstein, 1989*) analysis results were used to identify the nearest marker position to a detectable QTL region that surpasses the LOD threshold, which was determined in the permutation test at the GW level. Besides these two analyses, the Kruskal-Wallis (KW) test was also performed to detect significant marker-trait associations at $p < 0.05$.

The proportion of phenotypic variance explained (PVE) by a single QTL was calculated by the square of the partial correlation coefficient ($r^2$) using MapQTL ver. 6 (*Van Ooijen, 2009*). Based on the results of the QTL analysis, the genotype effects of significant QTLs for the traits were investigated using the analysis of variance (ANOVA) and t-test. Based on the physical position of the shortlisted markers on the EG5 genome, gene prediction was mined from MPOB's predicted gene list (http://genomsawit.mpob.gov.my).

## RESULTS

### SNP and InDel markers identification

A total of 2,684,179 raw reads were obtained. These showed variances from the reference genome with a minimum base quality confidence threshold of 25 and a minimum read depth coverage of five reads in each sample. These raw reads are comprised of 2,530,632 SNPs (94.3%) and 153,547 InDels (5.7%), with a file size of 1.6 GB and 110 MB sequence data, respectively.

For SNP marker discovery, 2,472,925 SNPs that had missing genotype calls of more than 10% across the progeny samples were excluded. The remaining 57,707 SNPs, which represented 2.33% of the total discovered SNP variances, were retained. For InDel markers, 3,854 markers were retained after the elimination process. Of these, 2,154 InDels (55.9%) fit into the genome build, and 1,700 InDels (44.1%) resided on the scaffold.

Among the 57,707 SNPs with a minimum genotype calls of 90%, there were 49,756 SNPs (86.2%) that were filtered out, and 7,951 informative SNPs were left. Ultimately, only 665 InDels (17.3%) were retained for the subsequent analysis. In total, 3,113 SNPs and 225 InDels were eliminated after evaluating by $x^2$ test for goodness-of-fit to the Mendelian segregation. As a result, 4,838 SNPs and 440 InDels were informative (Table S1) and were retained for the construction of the genetic linkage maps as shown in Table 1. Based on the results of the PCA analysis (Fig. 1), using a subset of 2,291 informative SNPs, it seems that the progenies belonged to the same cross.

### The *dura* (maternal) and *pisifera* (paternal) linkage maps

Among the 5,278 shortlisted markers, 3,224 SNPs (66.6%) and 300 InDels (68.2%) were heterozygous in the male parent, whereas 1,605 SNPs (33.2%) and 125 InDels (28.4%) were heterozygous in the female parent. Nine SNPs and 13 InDels showed the segregation patterns of being heterozygous in both parents with two alleles present. There were only two heterozygous InDels with three alleles. Based on the results, the heterozygosity markers were found to be more abundant in the paternal oil palm compared to the maternal oil palm (Table S2). Subsequently, the shortlisted markers, comprising 4,838 SNPs (91.7%) and 440 InDels (8.3%) were used to construct the genetic linkage map for Deli *dura* x Serdang *pisifera*. Of these, only 1,754 markers, consists of the segregation patterns lmxll, hkxhk and efxeg were used for the maternal map construction. Whereas 3,548 markers, comprising markers with the segregation patterns of nnxnp, hkxhk and efxeg were used for the paternal map construction. Of these, the redundant markers having identical genotype calls were excluded; 280 for the maternal and paternal map. In total, 236 markers for the maternal map and 109 markers for the paternal map were excluded due to of unsuccessful assignment to any grouping nodes.

Table 2 shows the markers that were assigned to the groups at a LOD score of 5.0 for the genetic linkage map. The major grouping nodes were confidently scored and assigned to 16 chromosomes corresponding with the physical map of the oil palm genome (*Singh et al., 2013*). Finally, 16 linkage groups for both parental maps were constructed using 1,239 markers for the maternal map (D200) and 2,918 markers for the paternal map (QP447),

**Table 1 Summary of the informative SNP and InDel markers of 112 progenies from a cross of Deli
*dura* and Serdang *pisifera*.** A total of 5,278 markers comprising 4,838 SNPs and 440 InDels were
retained for genetic linkage map construction. The italic font represents the markers that were excluded
in this analysis. Chr indicates the markers that reside on the EG5 genome build. The shortlisted SNP and
InDel markers were summarized in Table S1.

| Filtering parameters | SNPs | | InDels | | Total |
|---|---|---|---|---|---|
| | Chr. | Scaffolds | Chr. | Scaffolds | |
| **Total markers having <= 10% missing data** | 25,738 | 31,969 | 2,154 | 1,700 | 61,561 |
| *Total markers filtered out using Excel:* | *22,416* | *27,340* | *1,783* | *1,406* | *52,945* |
| *1) Major allele frequency >95%* | | | | | |
| *2) Major genotype frequency >95%* | | | | | |
| *3) Markers with genotypes that do not follow the Mendelian segregation patterns between progenies and parents* | | | | | |
| *Total markers filtered out using JoinMap4.1:* | *1,031* | *2,082* | *105* | *120* | *3,338* |
| *1) Segregation distortion level at p ≤ 0.001* | | | | | |
| *2) Identical loci with similarity value = 1.0* | | | | | |
| **Total markers remained after filtering** | 2,291 | 2,547 | 266 | 174 | 5,278 |

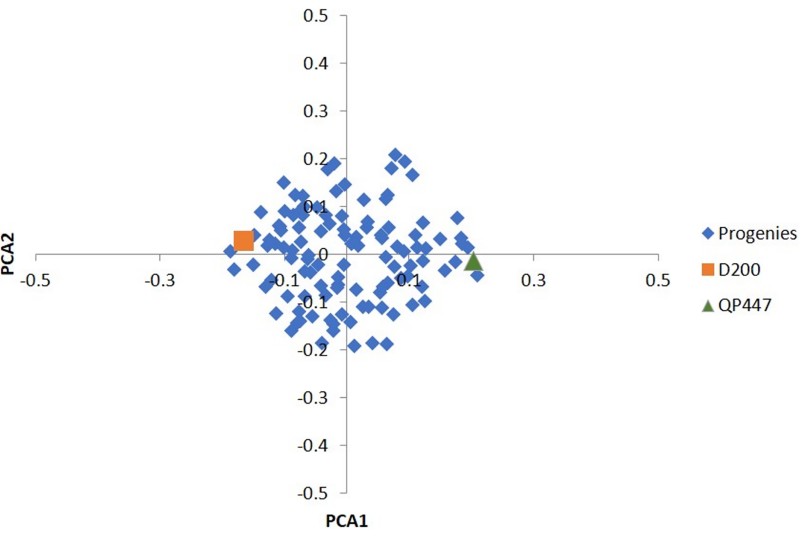

**Figure 1 PCA of the first two principal components (PCA1 *vs.* PCA2) for 112 progenies and their
parents.**                                                    

as shown in Table 3. The linkage groups were named according to the physical map (*Singh
et al., 2013*) for easy traceability. The LOD threshold was set at 5.0 to minimize erroneous
grouping assigned for each of the markers. The LOD value of 5.0 was the best synteny to
form the grouping with the chromosome from the reference genome based on the physical
map information.

   Furthermore, markers with a N.N. stress value of more than 4.0 were excluded in order
to retain only markers that were correctly assigned to the respective groups (*Van Ooijen,
2006*). The remaining 1,071 and 2,437 markers for the maternal and paternal maps,

**Table 2 Summary of the markers used for the construction of the genetic linkage map for parental lines of Deli *dura* and Serdang *pisifera*.**

| Linkage group (Maternal) | Number of markers | | | Linkage group (Paternal) | Number of markers | | |
|---|---|---|---|---|---|---|---|
| | Initial | Unanchored | Retained | | Initial | Unanchored | Retained |
| LG1 | 156 | 19 | 137 | LG1 | 176 | 26 | 150 |
| LG2 | 123 | 18 | 105 | LG2 | 255 | 58 | 197 |
| LG3 | 36 | 6 | 30 | LG3 | 336 | 56 | 280 |
| LG4 | 151 | 17 | 134 | LG4 | 204 | 26 | 178 |
| LG5 | 54 | 9 | 45 | LG5 | 382 | 59 | 323 |
| LG6 | 10 | 0 | 10 | LG6 | 262 | 29 | 233 |
| LG7 | 138 | 19 | 119 | LG7 | 146 | 29 | 117 |
| LG8 | 16 | 1 | 15 | LG8 | 264 | 58 | 206 |
| LG9 | 51 | 13 | 38 | LG9 | 134 | 20 | 114 |
| LG10 | 162 | 25 | 137 | LG10 | 60 | 5 | 55 |
| LG11 | 120 | 13 | 107 | LG11 | 114 | 21 | 93 |
| LG12 | 58 | 12 | 46 | LG12 | 125 | 16 | 109 |
| LG13 | 53 | 7 | 46 | LG13 | 135 | 30 | 105 |
| LG14 | 80 | 7 | 73 | LG14 | 104 | 19 | 85 |
| LG15 | 11 | 0 | 11 | LG15 | 160 | 24 | 136 |
| LG16 | 19 | 1 | 18 | LG16 | 61 | 5 | 56 |
| Total | 1,238 | 167 | 1,071 | Total | 2,918 | 481 | 2,437 |

**Note:**
The initial number of markers indicates the markers obtained after grouping marker at minimum LOD score of 5.0. Unanchored markers represent the number of markers that having N.N. Stress value of more than 4.0 and showing marker inconsistency compared to the physical map of EG5 genomes build.

**Table 3 Summary of Deli *dura* (D200) and Serdang *pisifera* (QP447) parental maps.**

| Linkage Group | Maternal (Deli *dura*, D200) | | | Paternal (Serdang *pisifera*, QP447) | | |
|---|---|---|---|---|---|---|
| | No. of markers | Length (cM) | Marker density (cM) | No. of markers | Length (cM) | Marker density (cM) |
| 1 | 137 | 290.14 | 2.1 | 150 | 303.91 | 2.0 |
| 2 | 105 | 338.65 | 3.2 | 197 | 517.50 | 2.6 |
| 3 | 30 | 118.92 | 4.0 | 280 | 385.00 | 1.4 |
| 4 | 134 | 292.18 | 2.2 | 178 | 328.06 | 1.8 |
| 5 | 45 | 150.39 | 3.3 | 323 | 462.45 | 1.4 |
| 6 | 10 | 17.48 | 1.7 | 233 | 405.61 | 1.7 |
| 7 | 119 | 259.81 | 2.2 | 117 | 221.57 | 1.9 |
| 8 | 15 | 40.92 | 2.7 | 206 | 401.65 | 1.9 |
| 9 | 38 | 158.42 | 4.2 | 114 | 287.35 | 2.5 |
| 10 | 137 | 227.81 | 1.7 | 55 | 151.83 | 2.8 |
| 11 | 107 | 336.81 | 3.1 | 93 | 183.52 | 2.0 |
| 12 | 46 | 130.26 | 2.8 | 109 | 244.73 | 2.2 |
| 13 | 46 | 108.37 | 2.4 | 105 | 232.98 | 2.2 |

| Linkage Group | Maternal (Deli *dura*, D200) | | | Paternal (Serdang *pisifera*, QP447) | | |
|---|---|---|---|---|---|---|
| | No. of markers | Length (cM) | Marker density (cM) | No. of markers | Length (cM) | Marker density (cM) |
| 14 | 73 | 159.03 | 2.2 | 85 | 170.31 | 2.0 |
| 15 | 11 | 57.34 | 5.2 | 136 | 130.50 | 1.0 |
| 16 | 18 | 51.02 | 2.8 | 56 | 144.62 | 2.6 |
| Total | 1,071 | 2,737.6 | – | 2,437 | 4,571.6 | – |
| Average | – | – | 2.9 | – | – | 2.0 |

**Note:**
The genetic linkage map will be used to facilitate the construction of QTL mapping. The maternal is denser than paternal map with the marker density of 2.9 cM and 2.0 cM, respectively.

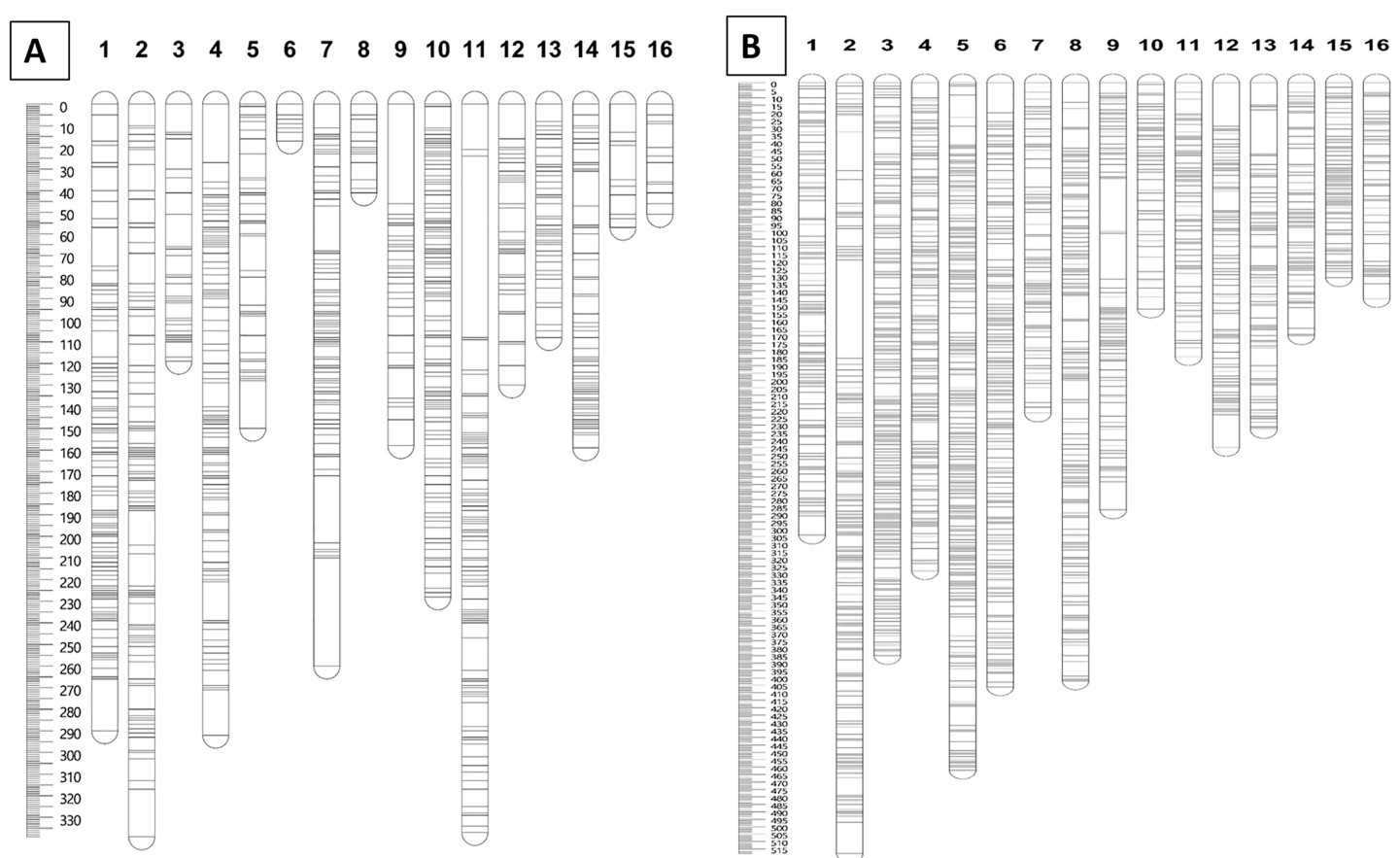

**Figure 2 Distribution of SNP and InDel markers on each of the linkage group for parental maps D200 (A) and QP447 (B).**

respectively, were fully informative in constructing the genetic linkage maps in this CP population. The marker distributions are presented in Fig. 2A for the maternal map (D200) and Fig. 2B for the paternal map (QP447). The maternal (D200, Deli *dura*) and paternal (QP447, Serdang *pisifera*) maps are 2,737.6 cM and 4,571.6 cM long, respectively.
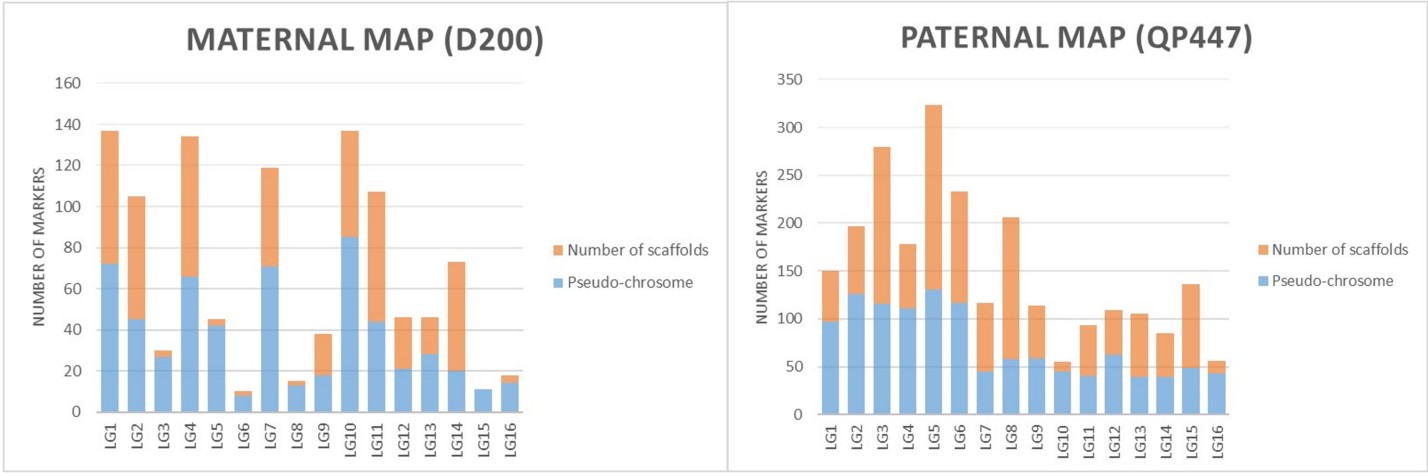

**Figure 3 The proportions of markers on the pseudo-chromosome and scaffolds throughout maternal and paternal maps.**

The average marker density for the maternal map is 2.9 cM; whereas for the paternal map, it is 2.0 cM. Likewise, the recombination rate for the maternal palm is 1.52 cM/Mb which is lower than the paternal palm's 2.54 cM/Mb.

The SNP and InDel markers used in the construction of the genetic linkage map were further investigated with respect to the physical map's position. The results in both maternal and paternal maps showed that the genetic linkage map would be able to position the markers on the scaffold to the respective linkage map as shown in Fig. 3.

## Verification of map orders

The collinearity of the markers in the genetic linkage and physical map was investigated as shown in Fig. 4. Only the markers that fell on the pseudo-chromosome were used in the investigation as these markers are located in the oil palm reference genome with the physical marker location (*Singh et al., 2013*). Figure 4 shows that the genetic map constructed in this study was highly correlated to the physical map as indicated by the R-squared ($r^2$) value, especially for linkage groups (LG) LG01, LG02, LG07, LG09, LG10, LG13 and LG14 for both the maternal and paternal maps with $r^2$ greater than 0.9. For maternal maps, all the groups showed high collinearity with an $r^2$ value of more than 0.7 between the linkage map and the physical map, except for LG06. Similarly, all the groups from the paternal map showed moderate to high collinearity, except for LG16 which showed very low collinearity with an $r^2$ value of 0.18.

## Distribution, correlation and heritability of the phenotypic data

In the Deli *dura* and Serdang *pisifera* cross, the coefficient of variations (CVs) of the phenotypic data were less than 30% for all nine analyzed traits, ranging from 2.2% to 23.3%. As shown in Table 4, the CVs were 17.7% for FFB, 20% for OY, 11.4% for O/B, 2.2% for O/DM, 7.2% for O/WM, 5.2% for M/F, 20.0% for K/F, 23.2% for S/F and 4.9% for F/B. Table 5 shows the results of the pairwise Pearson correlations and the significance levels. The OY and other traits, except for F/B, showed significant correlations above 0.80. On the

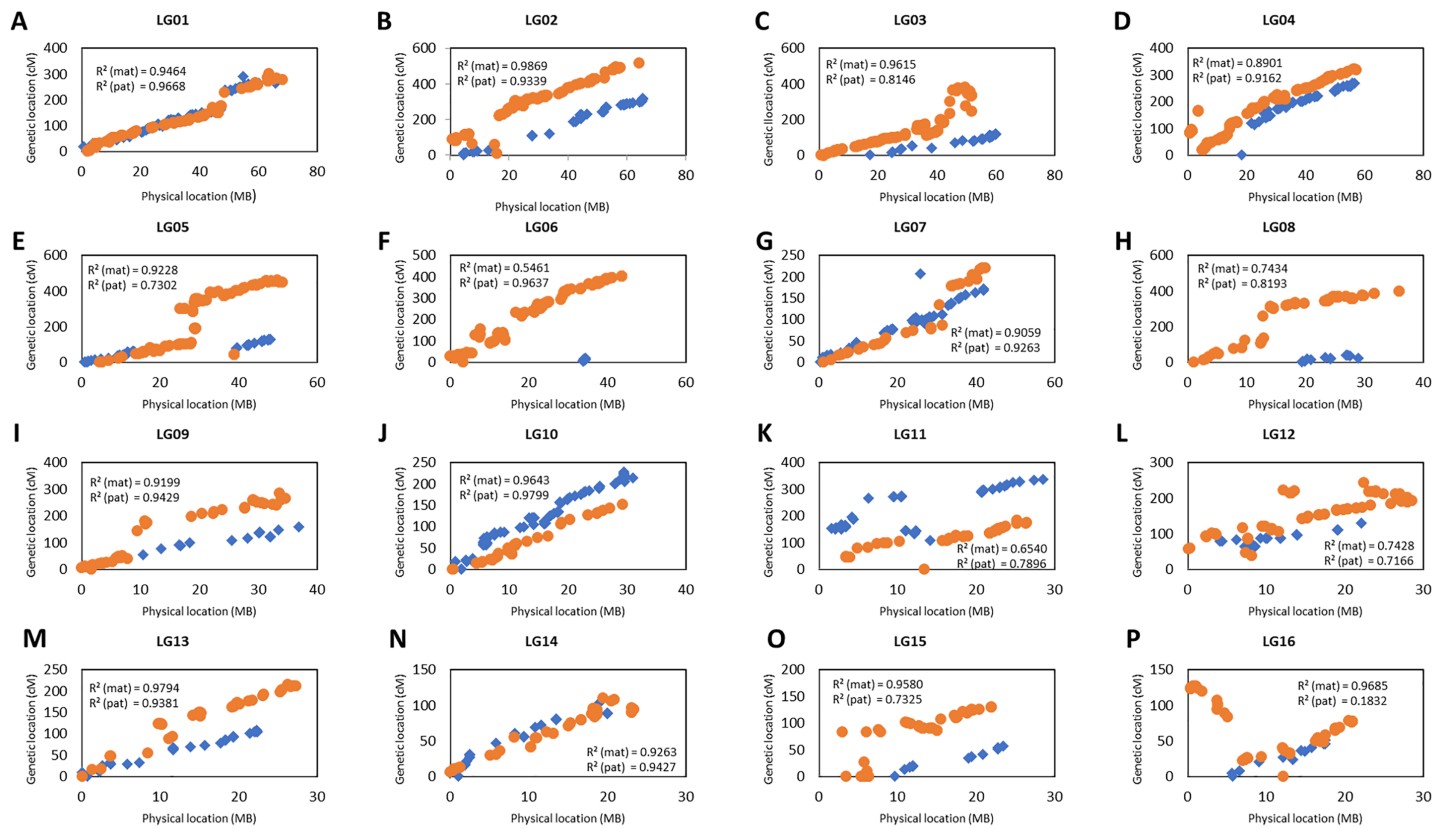

**Figure 4 Scatter plots of the marker positions in the genetic map and physical map throughout the oil palm genome.** Blue diamond represents the markers in the maternal genetic map, D200 and orange dots represent the paternal genetic map, QP447. Only markers reside on the 16 chromosomes are used to construct these scatter plots as reported by *Singh et al. (2013)*. The value of $R^2$ (mat, maternal map and pat, paternal map) indicates the correlation between genetic and physical maps.                               

**Table 4 Statistical analysis of the nine traits that were obtained from a cross of Deli *dura* and Serdang *pisifera*.**

| Trait | Abbreviation | Unit | n = 111 (For FFB) | | | | n = 97 (For bunch quality related data) | | | |
|---|---|---|---|---|---|---|---|---|---|---|
| | | | Mean | SD | CV (%) | Range | Mean | SD | CV (%) | Range |
| Fresh fruit bunch | FFB | kg/palm/year | 217.6 | 42.2 | 17.7 | 38.5–349.1 | – | – | – | – |
| Oil yield | OY | kg/palm/year | – | – | – | – | 58.5 | 11.7 | 20.0 | 28.2–89.7 |
| Oil-to-bunch ratio | O/B | % | – | – | – | – | 26.6 | 3.0 | 11.4 | 19.0–36.2 |
| Oil-to-dry mesocarp ratio | O/DM | % | – | – | – | – | 77.3 | 1.7 | 2.2 | 73.7–81.0 |
| Oil-to-wet mesocarp ratio | O/WM | % | – | – | – | – | 49.1 | 3.5 | 7.2 | 41.5–56.4 |
| Mesocarp-to-fruit ratio | M/F | % | – | – | – | – | 78.5 | 4.1 | 5.2 | 69.7–89.6 |
| Kernel-to-fruit ratio | K/F | % | – | – | – | – | 10.3 | 2.1 | 20.0 | 5.4–15.3 |
| Shell-to-fruit ratio | S/F | % | – | – | – | – | 11.0 | 2.6 | 23.2 | 4.9–17.9 |
| Fruit-to-bunch ratio | F/B | % | – | – | – | – | 68.9 | 3.4 | 4.9 | 57.5–78.2 |

**Note:**
The trait-of-interests are FFB, OY, O/B, O/DM, O/WM, M/F, K/F, S/F and F/B with n referring to sample size, SD referring to standard deviation and CV referring to coefficient of variation.

**Table 5 Pearson correlations among all the traits were calculated using the SPSS statistical software.**

|       | FFB | OY      | O/B      | O/DM     | O/WM     | M/F      | K/F       | S/F       | F/B      |
|-------|-----|---------|----------|----------|----------|----------|-----------|-----------|----------|
| FFB   |     | 0.819** | −0.131   | −0.061   | −0.079   | −0.028   | −0.058    | 0.086     | −0.165   |
| OY    |     |         | 0.454**  | 0.370**  | 0.359**  | 0.383**  | −0.382**  | −0.330**  | 0.117    |
| O/B   |     |         |          | 0.748**  | 0.760**  | 0.708**  | −0.578**  | −0.702**  | 0.457**  |
| O/DM  |     |         |          |          | 0.837**  | 0.367**  | −0.248*   | −0.432**  | 0.125    |
| O/WM  |     |         |          |          |          | 0.334**  | −0.257*   | −0.363**  | −0.033   |
| M/F   |     |         |          |          |          |          | −0.884**  | −0.898**  | 0.097    |
| K/F   |     |         |          |          |          |          |           | 0.617**   | −0.028   |
| S/F   |     |         |          |          |          |          |           |           | −0.148   |
| F/B   |     |         |          |          |          |          |           |           |          |

**Notes:**
* Correlation is significant at the 0.05 level (two-tailed).
** Correlation is significant at the 0.01 level (two-tailed).
Correlations with more than 0.5 are bold.

**Table 6 Heritability estimation of the trait of interests of *Tenera* (a cross of Deli *dura* and Serdang *pisifera*). $h^2$ refers to heritability estimation value (%).**

| Trait                    | Abbreviation | $h^2$ (%) |
|--------------------------|--------------|-----------|
| Fresh fruit bunch        | FFB          | 9.29      |
| Oil yield                | OY           | 0.47      |
| Oil-to-bunch ratio       | O/B          | 39.07     |
| Oil-to-dry mesocarp ratio | O/DM        | 2.89      |
| Oil-to-wet mesocarp ratio | O/WM        | 1.87      |
| Mesocarp-to-fruit ratio  | M/F          | 97.03     |
| Kernel-to-fruit ratio    | K/F          | 88.92     |
| Shell-to-fruit ratio     | S/F          | 83.96     |
| Fruit-to-bunch ratio     | F/B          | 18.31     |

other hand, K/F and S/F showing a strong negative correlation with M/F is not surprising. A similar correlation between M/F, and K/F and S/F has been reported in previous reports (*Ithnin et al., 2017*; *Jeennor & Volkaert, 2014*). Negative correlations were also observed between K/F and S/F and the oil factors, including OY, O/B, O/DM, and O/WM. Contrary to K/F and S/F, M/F showed a significant correlation with low to moderate positive correlations with the oil factors.

Heritability estimation values for the nine phenotypic traits are tabulated in Table 6. The heritability estimation ($h^2$) for FFB, OY, O/DM and O/WM was very low, ranging from 0.47 to 9.29%. The heritability value for F/B was slightly higher at 18.31%. In contrast, a moderate heritability estimation for O/B was recorded at 39.07%. In this population study, the estimation value for M/F was the highest at 97.03%, followed by K/F and S/F which were obtained at 88.92% and 83.96%, respectively.

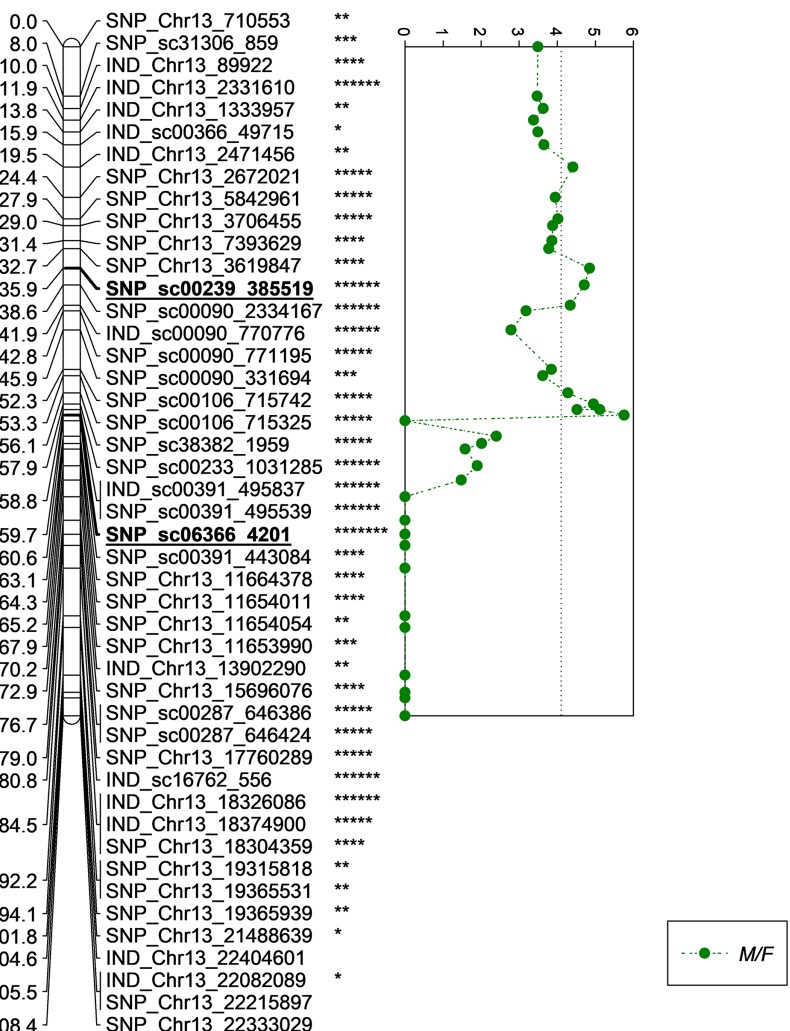

**Figure 5 Oil palm linkage map and distribution of QTLs associated with mesocarp to fruit.** The asterisk represents the significant value of KW test (*: 0.1, **: 0.05, ***: 0.01, ****: 0.005, *****: 0.001, ******: 0.0005, *******: 0.0001). The scatter plot illustrates the LOD scored obtained from IM and dashed line is the GW threshold of the permutation test.

## Identification of significant QTLs on the Deli *dura* (D200) and Serdang *pisifera* (QP447)

A total of eight significant markers associated with M/F, K/F and S/F were identified on LG13, which originated from the female parent (D200). Of these, two markers for M/F, five markers for K/F and one marker for S/F were discovered. All associated markers were located on chromosome 13 of the physical map (*Singh et al., 2013*).

One of the QTLs associated with M/F was located near SNP_sc00239_385519 with a phenotypic variation (PVE) of 20.6%. The SNP marker was located at 35.9 cM on the constructed genetic map and was located on the scaffold of the physical map (Fig. 5). Another QTL detected for M/F was detected on SNP_sc06366_4201 at 60.6 cM. This

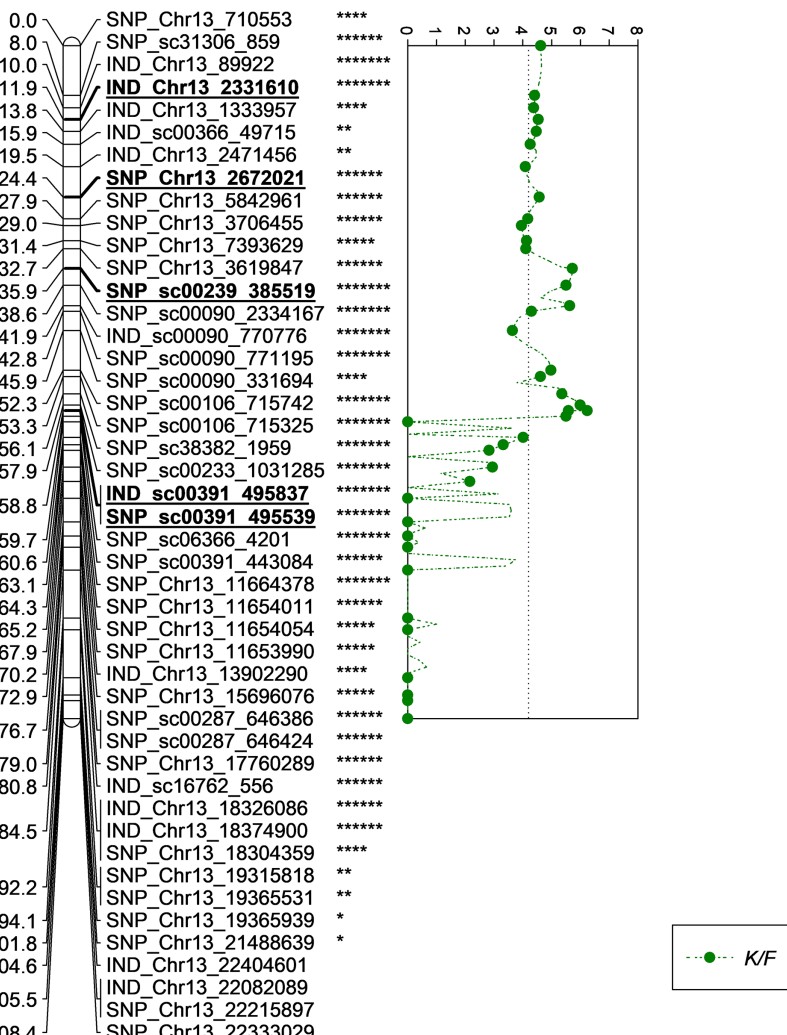

**Figure 6  Oil palm linkage map and distribution of QTLs associated with kernel to fruit.** The asterisk represents the significant value of KW test (*: 0.1, **: 0.05, ****: 0.005, *****: 0.001, ******: 0.0005, *******: 0.0001). The scatter plot illustrates the LOD scored obtained from IM and dashed line is the GW threshold of the permutation test.                   

marker explained 23.9% of PVE and is located on the scaffold of the physical map (Fig. 5). Another QTL peak was detected on the scatter plot, as shown in Fig. 5, at 19.5 cM, but this marker was excluded due to its insignificant value in the KW test.

Meanwhile, five other QTLs were identified for K/F at a GW significant threshold level of 4.1 as shown in Fig. 6. These markers gave a PVE value ranging from 19.4% to 25.6%. Out of the five markers, two of them were InDels namely IND_Chr13_2331610 and IND_sc00391_495837. The former was located on the chromosome while, the latter was positioned on the scaffold of the physical map. Other than InDels, three other SNP markers associated with K/F namely, SNP_Chr13_2672021, SNP_sc00239_385519 and SNP_sc00391_495539 were identified. Their PVE ranged between 19.5% and 25.6%.

**LG13_mat**

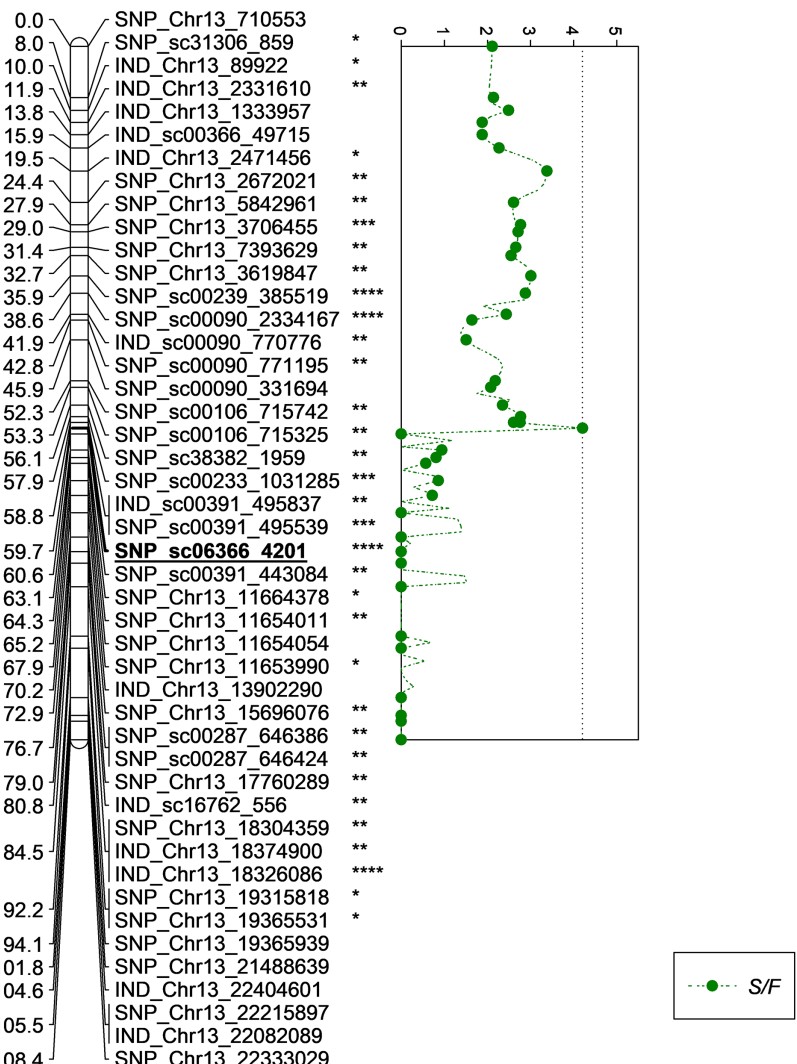

**Figure 7 Oil palm linkage map and distribution of QTLs associated with shell to fruit.** The asterisk represents the significant value of KW test (*: 0.1, **: 0.05, ***: 0.01, ****: 0.005). The scatter plot illustrates the LOD scored obtained from IM and dashed line is the GW threshold of the permutation test.               

Of these, only the first SNP was located on the chromosome and the rest were on the scaffold. The result showed that the QTLs for K/F, M/F and S/F were found on the maternal map instead of the paternal map. These traits could possibly be genetically controlled by the maternal line.

Besides QTLs for M/F and K/F, a QTL was also detected on SNP_sc06366_4201 for S/F at the GW's significant threshold level of 4.2 and PVE which was at 18.1% (Fig. 7). The SNP marker was located on the scaffold and at 60.6 cM in the genetic map. The QTL detected on S/F overlapped with K/F (Fig. 8). Unfortunately, there was no QTL region with a LOD value greater than the GW threshold identified for FFB, OY, O/B, O/DM, O/WM and F/B.

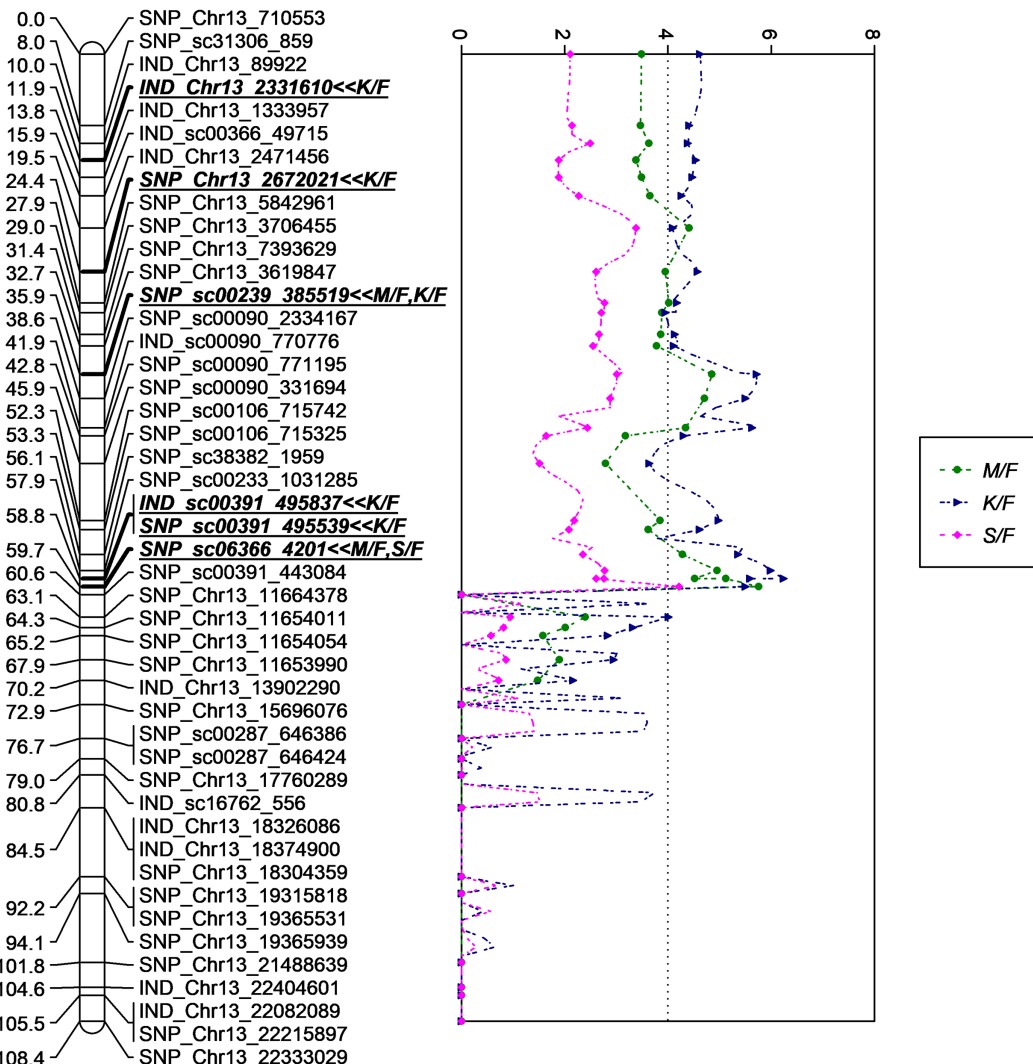

**Figure 8 Oil palm linkage map and distribution of QTLs associated with mesocarp-to-fruit kernel-to-fruit shell-to-fruit.**

In this study, the shortlisted markers influencing M/F, K/F and S/F were compared with the allele segregation pattern contributing to the traits, as shown in Fig. 9. In general, the shortlisted markers showed a significant contribution to the variation of the phenotypes as depicted in the statistical analysis (ANOVA and t-test) for all the traits. The mean M/F for oil palms with genotype AG was higher compared to those with genotype AA for SNP_sc00239_385519 (Fig. 9A). Nevertheless, palms with genotype AG of SNP_sc06366_4201 had a lower mean M/F than those with genotype GG (Fig. 9B). Palms carrying heterozygous genotypes for these two markers possessed a higher M/F than palms with homozygous genotypes.

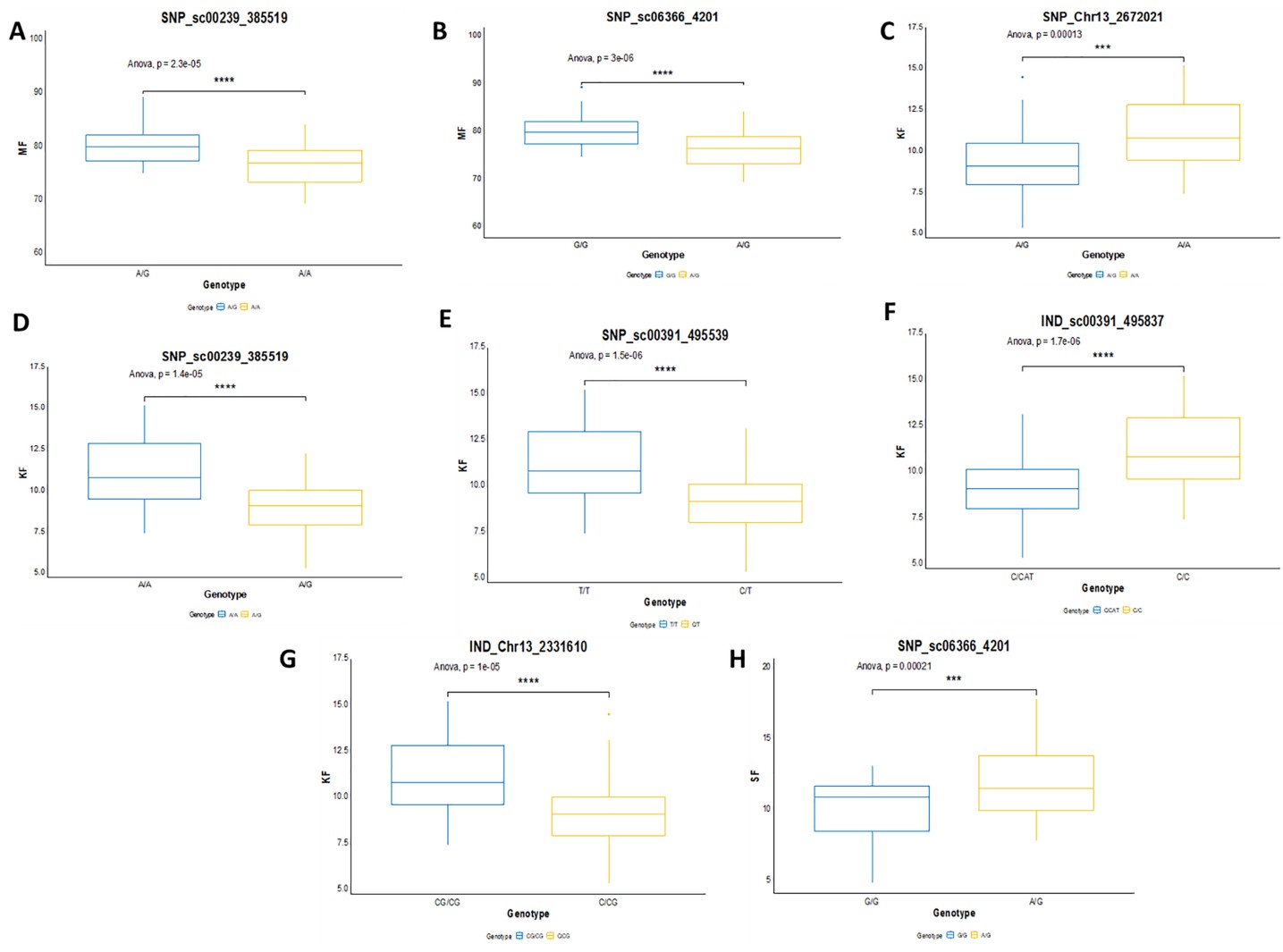

**Figure 9 Genotype effects of the significant markers.** (A and B) Effects on genotypes A/A and A/G of SNP_sc00239_385519 while G/G and A/G of SNP_sc06366_4201 on MF (mesocarp to fruit ratio). (C–G) Effects on genotypes A/A and A/G of SNP_Chr13_2672021 and SNP_sc0239_385519, while T/T and C/T of SNP_sc00391_495539 on KF (kernel to fruit ratio). Meanwhile, InDels' effects on C/C and CAT/C of IND_sc00391_495837 and C/CG and CG/CG of IND_Chr13_2331610 on K/F. (H) Effects on genotypes G/G and A/G of SNP_sc06366_4201 on SF (shell to fruit ratio). The asterisk represents significant *P*-value (***: $P \leq 0.001$, ****: $P \leq 0.0001$).

For K/F traits, palms with homozygous genotypes AA of SNP_Chr13_2672021, SNP_sc00239_385519 and TT of SNP_sc00391_495539 exhibited higher means of K/F than those with heterozygous genotypes (Fig. 9C–9E). For InDel markers, oil palms with genotypes C/C showed a higher mean K/F than those with genotypes CAT/C of IND_sc00391_495837 (Fig. 9F). Meanwhile, palms with genotype CG/CG showed a higher mean compared to those with genotypes C/CG (Fig. 9G). For S/F, a higher mean was observed when oil palms exhibited genotype AG compared to GG for SNP_sc06366_4201 (Fig. 9H). These effects on the mean phenotypes for genotype segregations are summarized in Table 7.

**Table 7 QTL effects of each genotype group in mean phenotypes of M/F, K/F and S/F of Deli *dura* (D200) and Serdang *pisifera* (QP447) cross.**

| Trait | Type of marker | Linkage Group | Chr[a] | Marker | Genotype | Mean (%) ± SD |
|-------|----------------|---------------|--------|--------|----------|---------------|
| M/F | SNP | 13 | scaffold | SNP_sc00239_385519 | AA | 76.802 ± 4.098 |
|  |  |  |  |  | AG | 80.503 ± 3.404 |
| M/F | SNP | 13 | scaffold | SNP_sc06366_4201 | GG | 80.402 ± 3.355 |
|  |  |  |  |  | AG | 76.616 ± 3.355 |
| K/F | SNP | 13 | scaffold | SNP_sc00239_385519 | AA | 11.137 ± 1.971 |
|  |  |  |  |  | AG | 9.300 ± 1.660 |
| K/F | SNP | 13 | 13 | SNP_Chr13_2672021 | AA | 11.031 ± 1.930 |
|  |  |  |  |  | AG | 9.435 ± 1.879 |
| K/F | SNP | 13 | scaffold | SNP_sc00391_495539 | TT | 11.233 ± 1.964 |
|  |  |  |  |  | CT | 9.307 ± 1.647 |
| K/F | InDel | 13 | scaffold | IND_sc00391_495837 | C/C | 11.233 ± 1.964 |
|  |  |  |  |  | CAT/C | 9.328 ± 1.668 |
| K/F | InDel | 13 | 13 | IND_Chr13_2331610 | C/CG | 9.329 ± 1.809 |
|  |  |  |  |  | CG/CG | 11.108 ± 1.915 |
| S/F | SNP | 13 | scaffold | SNP_sc06366_4201 | GG | 10.1109 ± 2.225 |
|  |  |  |  |  | AG | 12.008 ± 2.541 |

**Note:**
[a] Chromosome number as assigned by *Singh et al. (2013)*.

# DISCUSSION

In this study, the GBS approach was successfully utilized to discover an abundance of SNP and InDel markers in oil palm which has a genome size of approximately 1.8 Gb (*Singh et al., 2013*). The high stringency of inclusion markers with a 90% call rate was set up to minimize the false positives that could result from the sequence data analysis. Out of the retained SNPs, 25,738 SNPs (44.6%) were mapped onto MPOB's reference genome (EG5 genome build). The remaining 31,969 SNPs resided outside of the genome build, known as the scaffold that results from the incomplete assembly of the reference genome. As the reference genome was built based on the *pisifera* background, a detection of variance in the mapping population against the reference genome would be biased towards *pisifera*.

Besides that, the sequencing error was prone to a low read depth. To mitigate this issue, a few strategies had been taken to reduce erroneous results from the sequencing analysis. The strategies included filtering the obtained markers with high stringency, selecting a good pair of endonucleases and utilizing the oil palm reference genome. In this study, a pair of endonucleases, *Hind*III and *Taq*I produced high-quality variants in oil palm. *Hind*III, which consists of only a six base long sequence, is a frequent cutter-restriction enzyme and leaves a 4 bp overhang which serves efficiently for adapter ligation (*Elshire et al., 2011*). *Taq*I enzyme, on the other hand, is a dam methylation-sensitive enzyme that does not cut the methylated base of the recognition sequence which represses gene transcription (*Campbell & Kleckner, 1990*). Besides *Hind*III and *Taq*I. other restriction enzymes such as *Ape*KI (*Babu et al., 2020*), *Pst*I (*Osorio-Guarín et al., 2019*; *Bai et al.,*

*2018*), and the combination of *Pst*I and *Msp*I (*Pootakham et al., 2015*) had been applied in oil palm. The genotyping quality was also verified using the parental genotype information to rule out any erroneous Mendelian segregation of the progenies against their parental genotypes. A similar approach had been reported in bovine, whereby a minimum read depth of three with high genotyping quality stringency was applied (*Brouard et al., 2017*). To access the studied population relatedness, PCA results (Fig. 1) showed a tight cluster of all the progenies, indicating a closed relatedness within the population and originating from the same cross, which confirmed the progenies' genetic origin as biparental progenies.

In oil palm, the homozygosity level is expected to be higher in *dura* than in *pisifera* parents. This is due to the genetic background of *dura*, being less diverse and variable due to its narrow origin (*Corley & Tinker, 2016*). In its breeding history, *dura* came from only four different sources that were originally planted in Bogor, Indonesia (*Corley & Tinker, 2016*). Therefore, such observations were expected, and were also reported by *Ting et al. (2013)* in the Deli *dura* which appeared to be more homozygous than the Yangambi *pisifera*. In this study, similar findings were also discovered when the male parent, *pisifera*, had a double number of heterozygous SNPs and InDels compared to than the female parent, *dura*. The genetic maps are in agreement with the breeding history where the paternal map was observed to be denser than the maternal map, which explains the high heterozygosity of the paternal line generating higher map resolution.

The linkage map constructed in this study was able to position markers from scaffolds onto the respective chromosomes in the physical map, as shown in Fig. 3. The LG4 of the maternal map and LG5 of the paternal map, for example, showed the most abundant markers from the scaffolds that were able to be located on the chromosome. This might be due to the high recombination rate that occurred in the studied population. The recombination rate and distribution along chromosomes are varied across different populations, which potentially influences a population's genetic background and the breeding strategy that can be used (*Ma et al., 2001*; *Ong et al., 2019*). The linkage map reported in this study would be able to enhance the genome assembly of the oil palm described by *Singh et al. (2013)*. The enhancement of genome assembly through genetic linkage maps was also reported across kingdoms such as for *Daphnia magna* (*Dukić et al., 2016*), collared flycatcher (*Kawakami et al., 2014*) and soybean (*Lee et al., 2020*). Despite the total number of retained markers on the maternal map in this study being about 50% less than the number of markers on the paternal map, the markers were successfully grouped into 16 linkage groups on the maternal map. Having a larger number of markers in a genetic linkage map could dissect more linkage information that could be used in QTL analysis. Thus, increasing the marker size should be done in the future to obtain more genetic information about the oil palm. In order to capture more markers in the maternal line, increasing the number of samples and using different restriction enzyme combinations should be performed in the future.

In the present work, the total length of the genetic linkage map obtained is longer than those previously reported. *Pootakham et al. (2015)* constructed a genetic linkage map of 1,429.6 cM using 1,085 SNPs on Deli *dura* x Dumpy AVROS *pisifera*. Likewise, a genetic

**Table 8 Summary of the QTLs associated with M/F, K/F and S/F traits of Deli *dura* (D200) and Serdang *pisifera* (QP447) cross.**

| No. | Trait | GW[a] | CW[b] | QTL peak | QTL peak position in linkage map (cM) | PVE(%)[c] | Group | Nearest markers | KW[d] | Map |
|-----|-------|------|------|----------|----------------------------------------|-----------|-------|-----------------|-------|-----|
| 1 | M/F | 4.0 | 3.1 | 4.86 | 32.663 | 20.6 | LG13 | SNP_sc00239_385519* | 0.0005 | Maternal |
| 2 | M/F | 4.0 | 3.1 | 5.76 | 59.713 | 23.9 | LG13 | SNP_sc06366_4201 | 0.0005 | Maternal |
| 3 | K/F | 4.1 | 3.0 | 4.51 | 11.860 | 19.4 | LG13 | IND_Chr13_2331610 | 0.0001 | Maternal |
| 4 | K/F | 4.1 | 3.0 | 4.57 | 24.386 | 19.5 | LG13 | SNP_Chr13_2672021 | 0.0005 | Maternal |
| 5 | K/F | 4.1 | 3.0 | 5.72 | 35.948 | 23.8 | LG13 | SNP_sc00239_385519 | 0.0001 | Maternal |
| 6 | K/F | 4.1 | 3.0 | 6.24 | 58.811 | 25.6 | LG13 | SNP_sc00391_495539 | 0.0001 | Maternal |
| 7 | K/F | 4.1 | 3.0 | 6.24 | 58.811 | 25.6 | LG13 | IND_sc00391_495837 | 0.0001 | Maternal |
| 8 | S/F | 4.2 | 3.3 | 4.21 | 59.713 | 18.1 | LG13 | SNP_sc06366_4201 | 0.005 | Maternal |

Notes:
[a] Genome-wide LOD threshold.
[b] Chromosome-wide LOD threshold.
[c] Percentage of the phenotypic variance explained at the QTL.
[d] Significance level from Kruskal-wallis analysis.
* Nearest markers to the QTL peak.

linkage map of 1,527.0 cM was constructed by *Bai et al. (2017)* on Deli *dura* × Ghana *pisifera* using 1,357 SNPs and 123 SSRs. *Teh et al. (2020)* reported a linkage map of Gunung Melayu *dura* x Gunung Melayu *pisifera* spanning 1,618.51 cM that was constructed using 506 SNPs and 59 SSRs. Likewise, *Herrero et al. (2020)* reported a map length of 1,370 cM constructed using 2,388 markers on a cross of *dura* and Nigeria *pisifera*. As the numbers of SNP markers used in this study doubled those of the previous studies, verification with the physical map order was performed to investigate the compatibility of JoinMap software to handle a large number of markers. The marker order obtained in this study was highly correlated to both the maternal and paternal physical maps, with moderate to high collinearity $r^2$ values of more than 0.70. In contrast, the low collinearity results indicated by the $r^2$ value could be due to an insufficient number of informative markers (*Ong et al., 2019*). Therefore, obtaining more significant markers could address the problem. In the future, with the advancement of mapping software to handle a larger set of markers, the 90% number of missing calls used in this study could be lowered to 80%, as in *Bai et al. (2017)*. This could potentially lead to the discovery of more significant markers in the mapping population.

In this study, the stringency was set at the highest level possible in order to reduce sequencing error due to the low read-depth. The marker interval reported in this study is higher than the previous findings of 1.03 cM (*Bai et al., 2017*) and 1.30 cM (*Pootakham et al., 2015*) obtained using the restriction-site associated DNA sequencing (RAD-seq) approach. Although GBS and RAD-seq are both restriction-site sequencing approaches, GBS targets low-coverage sequencing compared to RAD-seq (*Beissinger et al., 2013*). Increasing target coverage sequencing should be able to reduce false-positive genotype calls, as the sequencing error is prone to short reads rather than long reads.

The significant threshold for both GW and CW was obtained at a *p*-value of 0.05 as shown in Table 8. However, only the GW threshold for each trait that showed a higher stringency than CW was selected for declaring significance QTL in each parental line.

The high stringency is necessary in order to avoid identifying the false QTL regions associated with any trait of interest. Obtaining the threshold based on the 1,000 iterations of the phenotype data would be able to increase the stringency of the data analysis. A total of eight significant markers associated with M/F, K/F and S/F were identified on LG13 of the maternal map and are located on chromosome 13 of the physical map (*Singh et al., 2013*). This could explain why the QTL regions found in this study would be specific to the Deli *dura* as no other findings had been reported for these three traits that had been identified on chromosome 13 (*Billotte et al., 2010*; *Jeennor & Volkaert, 2014*; *Teh et al., 2016*; *Babu et al., 2017*; *Ithnin et al., 2017*; *Teh et al., 2020*). Recently, *Babu et al. (2020)* discovered the significant regions associated with O/B and O/WM. As a similar reference genome developed by *Singh et al. (2013)* was used in this study, we compared and discovered that the reported region was located near the *SNP_sc00391_495539* and *IND_sc00391_495837* markers in this study with a distance of 32.14 kb and 31.85 kb, respectively. In this study, these two markers were associated with K/F. The results of this finding suggest that further research in this region such as fine mapping would be worthwhile.

The QTL for K/F found on the maternal map corresponded to that of the previous study by *Okwuagwu & Okolo (1992)*, which stated that K/F was generally inherited from the maternal side rather than the paternal side. The overlapping QTLs for K/F, M/F and S/F were also discovered by *Seng et al. (2016)*. Moreover, the identified QTLs for these three traits in this study were clustered and showed almost similar peak patterns in LG13 on the maternal map, as illustrated in Fig. 8. Based on the scatter plot, the sudden drop after the peak at 60.6 cM might be due to insufficient markers around that region (*Van Ooijen, 2009*). Therefore, adding more markers to the region might improve the resolution of QTL mapping. The shell thickness is governed by a single gene (*Beirnaert & Vanderweyen, 1941*) that could be less complex compared to FFB and oil yield related components. The lack of significant QTL identified for FFB and other oil yield-related components might be due to the fact thatthese traits are complex and controlled by multiple genes. In the future, as opposed to using a single QTL associated with a trait of interest, multiple QTL regions would be necessary to capture a larger portion of the genetic variance associated with a complex trait.

Heritability was observed to be a crucial factor for successful QTL identification. Traits with high heritability values such as M/F, K/F and S/F increases the efficiency of the QTL analysis.

FFB yield, OY, O/DM and O/WM showed low heritability values as they were largely controlled by environmental factors such as weather and harvesting practice (*Okoye, Okwuagwu & Uguru, 2009*). To overcome this drawback, prolonged recording of FFB data for five consecutive years could increase the heritability of a trait (*Corley & Tinker, 2016*). *Rafii et al. (2002)* also reported a very low heritability value for FFB which was less than 10%. A moderate to high value of narrow-sense heritabilities was reported for M/F and S/F with 43.94% and 39.38% for *pisifera*, respectively. On the other hand, low heritabilities for *dura* at 9.78% for M/F and 0.00% for S/F were reported by *Rafii et al. (2002)*. For K/F, moderate to low values of narrow-sense heritabilities of 22.81% for *pisifera* and 37.26% for

*dura* were observed. These findings agreed with *Menendez & Blaak (1964)*, who reported moderate to high narrow-sense heritabilities for M/F (80%), K/F (83%) and S/F (61%), indicating a major influence of genetics on these traits Therefore, heritability could explain the result of this study when the identified significant markers were highly associated with only K/F, M/F and S/F. Failing to discover significant markers associated with a trait with low to moderate heretabilities explained that the environment had more influence than genetics on a certain trait. Fruit formation, as an example, is highly dependent on the weevil population, which may affect the pollination rate, due to weather conditions and population abundance in different climatic conditions (*Prasetyo, Purba & Susanto, 2014*; *Teh et al., 2020*). Therefore, the failure to discover significant QTL for all traits, except for K/F, M/F and S/F in this study, might influence the traits' heritabilities. QTL regions showed a high association with these three traits, with the PVE ranging from 18.1 to 25.6% as summarized in Table 8.

In this study, LG 13 appeared to be the QTL 'hot spots', where all markers associated with K/F, M/F and S/F were located around this linkage group. Even though the study failed to unlock the potential QTL for the production traits such as FFB, OY, O/WM and O/WM, kernel and mesocarp traits, which are bunch component traits, are another major determining component for the oil palm yield production. Palm oil is extracted from two fruit components, the mesocarp and the kernel, for different purposes. Mesocarp is a major contributing factor to OY; meanwhile, kernel is for palm kernel oil. Clearly, this is a good indicator to improve oil palm yield production to drive national income. As previously discussed, the mesocarp is inversely proportionate to kernel size and shell thickness. Therefore, increasing the mesocarp size while decreasing the shell and kernel thickness could generate an oil palm with a higher oil yield. In 2021, Malaysia recorded RM82.49 billion (~USD18.64 billion) for the total export earnings of palm oil. The earnings could probably be increased by increasing mesocarp size for the oil extraction, where even a 1% difference could be translated into a RM0.83 billion gain or loss. Thus, screening the planting material carrying genotypes with high performance traits *via* MAS could result in high profitability.

In most of these cases, the peaks for other than K/F, M/F and S/F traits were still detectable. However, they were at the lower than significant GW and CW thresholds, causing less confidence to declare them as significant QTLs. It might be possible to detect QTLs associated with these traits by increasing the sample size and providing phenotypic data with more variations. Additionally, having more markers dispersed across the genome would also increase the detection power of these complex traits.

Most of the QTLs identified in the present study resided near the important gene functions controlling seed development and formation, as summarized in Table 9. Two out of eight significant QTLs appeared in two different traits, which may be pleiotropic effects of the gene. For example, *SNP_sc00239_385519*, associated with both K/F and M/F and positioned 86.2 kb away from Casein kinase II subunit alpha-2 isoform 1, is likely a candidate. This putative gene was suggested targeting for seed germination and seedling growth (*Mulekar et al., 2011*; *Vilela, Pagès & Riera, 2015*). Despite none of the identified QTLs being located in a genic region, the nearest position is less than 5 kb away from the

**Table 9 Putative genes detected nearby QTL regions associated with M/F, K/F and S/F.** None of marker resides within a genic region.

| Trait | Map | LG | Marker | Distance from the gene (kb) | Nearby putative gene | Putative function |
|---|---|---|---|---|---|---|
| M/F | Maternal | 13 | SNP_sc00239_385519 | 86.2 | Casein kinase II subunit alpha-2 isoform 1 | Seed germination and seedling growth (*Mulekar et al., 2011*; *Vilela, Pagès & Riera, 2015*) |
| M/F | Maternal | 13 | SNP_sc06366_4201 | NA | NA | NA |
| K/F | Maternal | 13 | SNP_sc00239_385519 | 86.2 | Casein kinase II subunit alpha-2 isoform 1 | Seed germination and seedling growth (*Mulekar et al., 2011*) |
| K/F | Maternal | 13 | SNP_Chr13_2672021 | 111.2 | Transcription factor bHLH18-like | Apical hook development (*Hao et al., 2021*) |
| K/F | Maternal | 13 | SNP_sc00391_495539 | 4.9 | Zinc finger CCCH domain-containing protein 18-like | Seed germination and embryo development (*Chen et al., 2020*) |
| K/F | Maternal | 13 | IND_sc00391_495837 | 5.2 | Zinc finger CCCH domain-containing protein 18-like | Seed germination and embryo development (*Chen et al., 2020*) |
| K/F | Maternal | 13 | IND_Chr13_2331610 | 28.6 | Inositol-tetrakisphosphate 1-kinase 2-like | Seed coat development (*Tang, Tan & Xue, 2013*) |
| S/F | Maternal | 13 | SNP_sc06366_4201 | NA | NA | NA |

**Note:**
NA: No available information

Zinc finger CCCH domain-containing protein 18-like, which controls seed germination and embryo development (*Chen et al., 2020*). Nevertheless, the present analysis was able to narrow down and reveal nearby predicted genes based on the physical position. Additionally, six shortlisted markers reside near genes (5 to 111 kb away) that are responsible for seed growth and plant development, such as seed germination, apical hook development and seed coat development. These QTLs may play a pivotal role in oil palm breeding programs as palm oil is extracted mainly from either the mesocarp or kernel parts. Nevertheless, further validation and fine mapping should be carried out to confirm and narrow down the genetic interval of the shortlisted QTLs.

In Malaysia, it was reported that the stagnancy of oil palm productivity might be due to various reasons, including unsuitable soils, poor estate management, and unskilled labor (*Jalani et al., 2002*). A yield gap was also observed between smallholders and plantations, where the plantations applied better field management practices compared to the smallholders. Thus, there is still room for improvement in order to close the gap in oil palm productivity, such as by providing high-quality seeds and implementing good estate management practices. The discovered QTLs associated with M/F, K/F and S/F in this study could be applied to screen and select palms at the nursery stage based on their genetic information. Selected palms with desired traits such as a thick mesocarp, a thin kernel and a thin shell could be identified prior to the planting stage. In addition, integrating the discovered QTLs could shorten the time taken to release new oil palm varieties. Therefore, this study is very useful for the oil palm industry.

## CONCLUSION

High-density linkage maps spanning 2,737.6 cM for the maternal and 4,571.6 cM for the paternal with an average marker density of 2.9 cM and 2.0 cM, respectively, were developed using 5,278 markers. This study successfully positioned 2,721 markers on the scaffold relative to the respective linkage maps, thus contributing to a noticeable improvement in the oil palm's genome assembly. The genetic linkage maps demonstrated high level of collinearity with their corresponding physical maps. Seven QTLs were identified with two for M/F, four for K/F and one for S/F. These are fruit component traits that have high heritability values. All significant QTLs discovered at chromosome 13 on the maternal map suggest that these traits are genetically controlled by the maternal line. Candidate gene analyses of the QTLs for K/F, M/F and S/F identified the QTL regions located nearby the gene, controlling seed development and formation.

The discovered markers in this study, however, require further validation to investigate their robustness through fine-mapping. In the future, generating $F_2$ populations would produce more variation in phenotypic data and allelic segregation.This would allow for better coverage for the construction of a genetic linkage map and QTL analysis. In order to increase the QTL detection power, increasing the sample size should be one of the possible approaches. Nevertheless, the QTLs detected in this study are population-specific to the Deli *dura* and Serdang *pisifera* cross; thus, further verification should be worthwhile when different genetic backgrounds are selected.

## ACKNOWLEDGEMENTS

The authors would like to thank Lee Weng Wah and Tuan Noorasyikin Tuan Man of GLS Sdn. Bhd. (Malaysia) for their valuable advice on bioinformatics. Many thanks to Chua Kia Ling and Tay Chee Chun (GAT Sdn Bhd, Malaysia) for sharing knowledge on the oil palm breeding perspective. We would also like to thank all the reviewers for their constructive comments and suggestions on this article.

### Funding

This study was financially supported by Genting Plantations Berhad, ACGT Sdn. Bhd, Universiti Putra Malaysia and the Department of Agriculture Sabah, Malaysia.
The funders had no role in study design, data collection and analysis, decision to publish, or preparation of the manuscript.

### Grant Disclosures

The following grant information was disclosed by the authors:
Genting Plantations Berhad, ACGT Sdn. Bhd, Universiti Putra Malaysia.
Department of Agriculture Sabah, Malaysia.

## Competing Interests

Fakhrur Razi Mohd Shaha, Pui Ling Liew, Hui Yee Yong and Soo Heong Boon are employed by ACGT Sdn. Bhd. Jakim Barin and Justina Rolland are employees of the Department of Agriculture Sabah.

## Author Contributions

- Fakhrur Razi Mohd Shaha conceived and designed the experiments, performed the experiments, analyzed the data, prepared figures and/or tables, authored or reviewed drafts of the article, and approved the final draft.
- Pui Ling Liew conceived and designed the experiments, analyzed the data, prepared figures and/or tables, authored or reviewed drafts of the article, and approved the final draft.
- Faridah Qamaruz Zaman analyzed the data, authored or reviewed drafts of the article, and approved the final draft.
- Rosimah Nulit analyzed the data, authored or reviewed drafts of the article, and approved the final draft.
- Jakim Barin performed the experiments, authored or reviewed drafts of the article, performed breeding trial, trait recording and supplied plant materials, and approved the final draft.
- Justina Rolland performed the experiments, authored or reviewed drafts of the article, performed breeding trial, trait recording and supplied plant materials, and approved the final draft.
- Hui Yee Yong conceived and designed the experiments, analyzed the data, authored or reviewed drafts of the article, and approved the final draft.
- Soo Heong Boon conceived and designed the experiments, analyzed the data, authored or reviewed drafts of the article, and approved the final draft.

## Data Availability

The raw data is available in the Supplemental Files.

## Supplemental Information

Supplemental information for this article can be found online at http://dx.doi.org/10.7717/peerj.16570#supplemental-information.

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
