# Peer review of "Genotyping by sequencing for the construction of oil palm (Elaeis guineensis Jacq.) genetic linkage map and mapping of yield related quantitative trait loci"

_PeerJ, doi:10.7717/peerj.16570_

## Round 0.1 · original submission · Major Revisions

Please address the comments raised by the reviewers. The scientific language of the manuscript should be improved. The authors should attempt to build an integrated genetic map for the progeny. Moreover, the authors should present the heritability estimate of the phenotypic traits. It is also required to identify the underlying genes of the identified SNPs that may take part in the control of the studied trait.

Reviewer 1 ·

Basic reporting

In this manuscript author stated that, there was no significant marker discovery to identify QTLs of all nine investigated traits, nonetheless this study successfully discovered markers associated with K/F, S/F and M/F. However, overall very scientific work and clear presentation. Topic is relevant for the current scenarios as oil palm breeding for oil production. There are some points and basics that must be addressed at major level.
1. Major comment is without heritability estimates in phenotypic traits the study do not make sense.
2. At the moment for publication, it is not enough just to describe the SNP linked to QTL. It is required via SNP to identify the genes that take part at the control of the studied trait. This will also allow to confirm the connection of the selected gene loci marked by SNP in the control of the studied traits.
I can not comment on these before knowing the heritability.

According to my opinion, adding the above information, a new version of the article will be fit for publication in PeerJ.

Experimental design

.

Validity of the findings

.

·

Basic reporting

The writing is clear and understandable. However, there are some grammatical mistakes, which need to review and edit.
It is suggested to merge the figures with the same context or concept in one with sub-sections as A, B, etc.
The Tables with not the key results may move to supplementary files. As the information of Table-2 has been repeated in text and in the table, hence it can delete or add to the supplementary file.

Experimental design

Experimental design and methodology are well established. For some fixed standards, it is suggested to add the references of older studies.

Validity of the findings

Results are reproducible. however the conclusion is not well structured, may need to re-write.

Additional comments

Comments has been added in draft

Reviewer 3 ·

Basic reporting

The usage of english language is fine but the way of reporting sometimes contained statement which is quite confusing (comments highlighted in the manuscript) and need to be rephrased (Introduction, MM
& Discussion sections). Most references/citations needed to support the author's view of the discussion rather than putting statement which might be not true.

Too many sentences started with "A total of ...", and the sentences were too descriptive, thus hard to follow (in Result section).

Experimental design

still not getting the reason why to build genetic maps separately for paternal and maternal side, while the author should also attempt to build an integrated genetic for the progeny.

Validity of the findings

Validation for genetic map against physical genome build is fine.
Validation of the genotypes here must complete with statistical analysis (ANOVA),

Additional comments

Dear author,
The experiment design for your project is fine but there're many statements (highlighted and commented in the manuscript-pdf version) need to be clarified further or cross referenced to other journals for proper justifications.

Annotated reviews are not available for download in order to protect the identity of reviewers who chose to remain anonymous.

---

## Round 0.2 · Major Revisions

The authors should elaborate on their results in a more scientific way, and discuss how the findings could be impactful and significant to the readers. The authors must cite the relevant and recent literature that seems missing in the manuscript. Some comments are mentioned in the annotated file from reviewer 3.

Reviewer 1 ·

Basic reporting

Author has addressed all the questions which were asked and made changes accordingly. Now the manuscript has potential for publication

Experimental design

Author has addressed all the questions which were asked and made changes accordingly. Now the manuscript has potential for publication

Validity of the findings

Author has addressed all the questions which were asked and made changes accordingly. Now the manuscript has potential for publication

Additional comments

Author has addressed all the questions which were asked and made changes accordingly. Now the manuscript has potential for publication

·

Basic reporting

the manuscript has been well-revised for linguitics, references citations tables, and figures.

Experimental design

Experimental design is satisfactory

Validity of the findings

Results are reproducible.

Reviewer 3 ·

Basic reporting

The overall reporting here is clear and straightforward.

Though I find the literature references could be enhanced further as some statements need to be further elaborated (marked in the manuscript).

The quality and contents for both figures and tables are satisfactory.

However, there seems to be no new elements/messages to be delivered to audience from this study, perhaps author could further spell out the significant or impact from this study.

Experimental design

This is quite a standard experimental design. The markers filtering steps were quite robust and applied to the standard practice. As mentioned in the first review, it is possible to obtained sex-averaged genetic maps from both parental maps.

Validity of the findings

I gathered the findings from this study was identified markers associated to S/F, M/F and K/F (from maternal map only, how about paternal map?). However, from the earlier comments and reported by author, those traits were highly correlated. Hence the associated markers found in this study could be contributive to all or either one of the trait. No specific conclusion could be drawn from here unless further studies to conducted, such as fine mapping.

Additional comments

There are couple statements need to be addressed by author for clarity (marked in the manuscript). In general, the discussion was written in brief manner. The findings and impact from this study was not mentioned, like how to contribute to breeding improvement in oil palm.

Annotated reviews are not available for download in order to protect the identity of reviewers who chose to remain anonymous.

---

## Round 0.3 · Major Revisions

The authors have addressed some of the comments and queries raised by the reviewers. The manuscript still needs revision as per reviewer three comments.

Reviewer 3 ·

Basic reporting

no hypothesis was stated in the study.

overall reporting improved after 2 rounds of revision but certain statements (stated below) to be elaborated further for clarities.

the language needs to be improved further. Suggest author send for professional language editing. In particular, the words "successfully", "nevertheless" were repeated many times.

Experimental design

OK

Validity of the findings

Conclusions were not well stated but rather generalized

Additional comments

Certain statements are quite general although to citations. Authors need to elaborate further or link to the aim of you study here.

Annotated reviews are not available for download in order to protect the identity of reviewers who chose to remain anonymous.

---

## Round 0.4 · accepted · Accept

The authors have responded to all the queries and paper is now acceptable in PeerJ

The Section Editor noted:

> Minor changes are required. The wording of the methods is a little ambiguous. Was MapQTL used for the interval mapping and KW analysis? As written it only seems to be used for PVE. Also, Lander and Botstein's paper should be cited for interval mapping (Lander and Botstein, Genetics, 1989)